# Efficacy of Different Testing Scenarios in Reducing Transfusion-Transmitted Hepatitis B Virus (TT-HBV) Infection Risk

**DOI:** 10.3390/v14102263

**Published:** 2022-10-15

**Authors:** Nico Lelie, Michael Busch, Steven Kleinman

**Affiliations:** 1Lelie Research, 1811 DK Alkmaar, The Netherlands; 2Vitalant Research Institute, San Francisco, CA 94105, USA; 3Department of Pathology, University of British Columbia, Victoria, BC V6T 1Z4, Canada

**Keywords:** HBV transmission, efficacy, residual risk, screening scenarios

## Abstract

The efficacy of different screening scenarios in reducing hepatitis B virus (HBV) transmission risk as compared to the risk without screening was modeled in 9,337,110 donations from four geographical regions that had been subjected to hepatitis B surface antigen (HBsAg) and individual donation nucleic acid amplification testing (ID-NAT). We used the Weusten models for estimating infectivity risk for Red Blood Cell (RBC) transfusions in eight HBV infection stages and then evaluated multiple screening strategies based on minipool (MP) and ID-NAT options of different sensitivity for their efficacy in reducing this risk. The efficacy in reducing HBV transmission risk by screening scenarios across the regions varied between 81% (HBsAg only) and 99.2% (ID-NAT and anti-HBc). Highly sensitive ID-NAT alone achieved a slightly higher risk reduction (97.6–99.0%) than minipool of 6 donations (MP6)-NAT in combination with HBsAg and anti-HBc (96.3–98.7%). In ID-NAT screened lapsed and repeat donors, the additional risk removed by HBsAg testing was minimal (<0.1%). The modeling outcomes in this and two previous reports using this multi-regional database suggest that one could consider an ID-NAT alone testing scenario as an alternative to MP-NAT and serology-based testing algorithms and restrict serologic testing to first-time donors only.

## 1. Introduction

We previously compared the efficacy of various serologic and NAT testing options in reducing transmission risk for human immunodeficiency virus (HIV) and hepatitis C virus (HCV) by assessing screening data on 11 million donations from 22 blood establishments in six geographical regions [1,2]. Although similar data were collected for HBV, additional analyses and modeling work was deemed necessary prior to performing a similar efficacy analysis. As a first step, we developed a multi-stage classification system for HBV infection depending on the HBV marker detection profiles and published the rates of detection in these different stages in donations from first-time, lapsed and repeat donors [3]. Secondly, we refined models for estimating the residual risk of HBV window period (WP) infection [4,5] by adjusting values of input parameters for the occurrence of acute occult infections (i.e., HBsAg remaining nonreactive in acute phase), anti-HBs ‘vaccine breakthrough’ infections and abortive infections [6]. Thirdly, since the NAT assay (Ultrio) used in this multiregional study had been subsequently replaced by a more sensitive assay (Ultrio Plus), it was important to adjust our calculations to reflect the expected improved performance of this as well as newer assays (Ultrio Elite, cobas MPX) to make our analysis pertinent to contemporary approaches to HBV screening [7,8,9]. Fourthly, another infectivity model by Weusten and colleagues became available for estimating transmission risk from donations given by donors with occult HBV infection (OBI) [10], for which we first had to confirm the quantitative estimate of the 50% infectious dose (ID_50_) by comparing the modeling outcomes in this multi-regional study with the observed transmission rates in several lookback studies [9]. Finally, we investigated the very low viral load distribution and related TT-HBV risk in HBsAg-positive donors without detectable HBV-DNA to understand the safety of a theoretical scenario in which their donations would be used for transfusion [11]. Based on the above published analyses, we are now able to evaluate infectivity risk over the full spectrum of HBV infection stages and to produce a reliable efficacy analysis using the multi-regional ID-NAT screening database. 

In the present manuscript, we model the residual risk of different serologic and NAT testing strategies using the ID-NAT and HBsAg screening data from four of the six geographical regions of the multi-center study, i.e., South Africa, South-East (SE) Asia, Mediterranean Europe and Central-North (CN) Europe. We chose to analyze data from these four geographical regions because the number of donations (9.3 million in total) and ID-NAT yield rates were judged to be high enough to allow for estimation of the residual WP and OBI transmission risk. In each of these regions, we quantified the TT-HBV risk for eight different infection stages and calculated the overall risk if no HBV screening had been performed. This allowed for expressing the residual risk in each infection stage as a percentage of the overall risk without HBV screening. In the next step, we calculated the TT-HBV risk in the eight infection stages for different HBV screening strategies, i.e., serologic HBsAg screening with and without anti-HBc testing, multiplex ID-NAT alone with either one of two widely used automated molecular test systems and the combined approach of using both serologic testing and NAT methods in either ID, MP6-, or MP16-NAT format. This enabled comparing the efficacy of twelve testing scenarios in reducing TT-HBV risk using the ID-NAT and HBsAg screening data from the four evaluated geographical regions. 

## 2. Methods

### 2.1. Screening Data

Between 2005 and 2011, we collected HBV ID-NAT and HBsAg screening and supplementary testing data from donors of different donation categories, i.e., first-time, lapsed and repeat donations (as well as grouping the lapsed plus repeat donation data and reporting data for all donations in aggregate), from 22 blood establishments in six geographical regions [3]. Most of these establishments provided data from the first 2–5 years after the introduction of HBV-NAT screening, but some did not include the first year(s) of HBV-NAT results in their datasets. The present analysis of the multi-center study data includes 3,571,315 donations from South Africa, 1,051,381 from South East Asia, 2,096,732 from Mediterranean and 2,617,692 from Central–North Europe. The data from Egypt and Oceania (80,631 and 1,542,480 donations, respectively) were not included because the number of WP-NAT yield cases was judged to be too low for reliable projections of residual risk.

### 2.2. ID-NAT Yield Rate Enhancement Factors

The residual HBV WP and OBI transmission risk estimates of six geographical re-gions of this multi-center study were recently published for the Ultrio Plus assay in a separate paper [9]. That article describes how the HBV ID-NAT and HBsAg PRISM screening data from South Africa [7] and Hong Kong [8] were used to estimate Ultrio Plus over Ultrio NAT yield enhancement factors and how these would impact the prevalence rates in different infection stages and donation categories (data summarized in Appendix A). Specifically, we observed that the published analytical sensitivity values for the previous Ultrio assay could not be applied in the Weusten risk models because the assay under-detected a significant proportion of HBV infected donors, probably due to variation in the double stranded portion of the HBV genome [7,8]. This conclusion arose from our observations that after introduction of the more sensitive Ultrio Plus assay, there was an unexpectedly large increase in NAT yield rates, both for HBV WP infection (1.6–2.7 fold) and for OBI (1.7–3.0 fold) after correction for the HBsAg prevalence in sequential screening periods (Appendix A). The prevalence rates of the different infection stages in South-East Asia were predicted using the adjustment factors found for mainly HBV genotype B and C infections in Hong Kong, whereas the adjustment factors found in South Africa for mainly HBV genotype A were not only applied to the South-African screening data but also to those from Mediterranean Europe and Central-North Europe (Appendix A). We choose to use the South-African adjustment factors for these latter regions because these were the lowest (approximately 1.7-fold) and HBV genotype A is prevalent in South Africa as well as in Europe, although in the Mediterranean region HBV genotype D infections are also prevalent.

### 2.3. Residual Risk and Efficacy Analysis of Different Testing Scenarios 

An accurate analysis of efficacy in risk reduction for different HBV screening strategies is a complex task and depends on multiple factors. Figure 1 explains the foundations of a reliable efficacy analysis, and Figure 2 illustrates the impact of different HBV screening assays on risk reduction. Figure 3 summarizes the subsequent modelling steps for estimating the efficacy of different testing scenarios, each of which is documented by a table in the present report (i.e., Appendix A and Table 1, Table 2, Table 3 and Table 4).

In order to estimate the efficacy of different HBV screening scenarios, we first estimated the transmission risk in the absence of any HBV screening in different infection stages (as characterized by different HBV marker detection profiles) using the ID-NAT screening data on first-time, lapsed and repeat donations of the four geographical regions. In Figure 2, eight stages of HBV infection (a, b, c, d, e, f, g and h) are indicated for an Ultrio Plus ID-NAT and serology screening strategy. The parameters a, e and g represent residual risk/million donations and the parameters b, c, d, f and g represent yield rates/million donations (Figure 2 and Table 1). 

As previously described, we used the Weusten infectivity models [4,10] for the TT-HBV risk modeling in the WP and OBI stages in this multi-center study [9]. The ID_50_ is an important driver of the residual risk in these infection stages and was estimated at 3.16 (between 1 and 10) virions for the anti-HBc-negative infection stages and 100-fold higher with an estimate of 316 (between 100 and 1000) virions for the later anti-HBc-positive stages. The validity of these ID_50_ values has been established in two review papers, one comparing the modeled residual risk against observed TT-HBV infection rates in several lookback studies [9] and the other comparing infectivity data in different HBsAg-positive infection stages [11]. For the residual risk calculations, we used previously reported 95% and 50% lower limits of detection (LOD)s of 41.2 and 4.1 copies/mL for Ultrio Plus [12] and 7.5 and 1.6 copies/mL for cobas MPX assays, respectively [13].

In Table 4 of the recently published TT-HBV review article [9] (reproduced as Appendix A in the present paper), we used the Weusten WP and OBI risk models to calculate residual risk for different blood components/plasma transfusion volumes and for various ID-NAT and MP-NAT testing options and expressed this as a percentage of the Ultrio Plus ID-NAT yield rate. For the efficacy analysis in the present paper, we used the same TT-HBV risk conversion factors but reported them only for RBC transfusions containing an assumed transfusion plasma volume of 20 mL. Table 1 gives these infectivity-conversion factors for the eight infection stages in an Ultrio Plus ID-NAT and HBsAg PRISM screening scenario as well as for other ID and MP-NAT screening options. For example, with the scenario of Ultrio Plus ID-NAT and PRISM HBsAg screening, the following percentages were calculated to transform the yield rates to TT-HBV risk in the eight infection stages:Pre-ID-NAT WP: Pre-HBsAg ID-NAT yield rate (b) × 56.5% [4,5,6].Pre-HBsAg ID-NAT yield detection period: yield rate (b) × 100% [4,5,6].HBsAg+/HBV-DNA + stages: yield rate (c) × 100% [4].Post-HBsAg ID-NAT yield detection period: yield rate (d) × 30% [9,10].Post-ID-NAT WP: Post-HBsAg ID-NAT yield rate (d) × 5.2% [9,10].HBsAg+/HBV-DNA-nonreactive: HBsAg+/HBV-DNA- yield rate (f) × 6% [10,11].OBI ID-NAT yield: OBI ID-NAT yield rate (g) × 30% [9,10].OBI ID-NAT-nonreactive: OBI ID-NAT yield rate (g) × 5.2% [9,10].

For stage a, the pre-ID-NAT WP yield infectivity conversion factor was 56.5% (i.e., 13.1 pre-ID-NAT WP risk days divided by a 23.1 day HBsAg-negative, HBV-DNA-positive detection period as published) [6,7]. For the HBV-DNA-positive, pre-HBsAg detection period (stage b), we assumed 100% infectivity risk per RBC transfusion. We made this simplifying assumption even though some early WP donations may not be infectious, for example if neutralizing anti-HBs is present in vaccine breakthrough infections [6]. Additionally, we assumed 100% infectivity risk in stage c—the HBsAg and HBV-DNA-positive stage—even though at the end of this stage when anti-HBc is present and HBsAg and HBV-DNA are declining the probability of infectivity may be less than 100% [11]. The infectivity conversion factor in stages g and h were estimated using the Weusten OBI viral load distribution-risk model [10] that also can be applied to the IgM anti-HBc-positive late acute phase (stages d and e). The model predicts 30% probability of infectivity of HBV-DNA and anti-HBc (or IgM-anti-HBc)-positive but HBsAg-negative donations (stages d and g) [9]. The probability of infectivity reduces to 5.2% in the post-HBsAg and OBI infection stages that are ID-NAT (Ultrio Plus)-nonreactive (stages e and h, respectively) [9]. The Weusten OBI risk model [10] has also been applied to the viral load distribution in HBsAg and anti-HBc-positive donations that are ID-NAT-nonreactive (stage f). The model predicts that 6% (5–7)% of HBsAg-positive but Ultrio Plus-nonreactive RBC transfusions would be infectious if HBsAg and anti-HBc testing was not performed. In earlier publications [10,11], this percentage was estimated to be 9 and 15% using Egyptian and South African data, respectively, but after reassessing the available data for the Ultrio Plus assay, we estimated the percentage to be between 5 and 7% by using two approaches [4,10] on two datasets. Appendix A with an erratum gives the details of the residual risk analysis on the two datasets supporting that 6% infectivity risk of Ultrio Plus HBV-DNA-nonreactive HBsAg-positive donations is a reasonable assumption for RBC transfusions. 

Similarly, we used the Weusten WP and OBI models [4,10] for calculating infectivity risk conversion factors in each of the eight HBV infection stages for the other evaluated NAT screening options, i.e., MP16-NAT for the Ultrio Plus assay and ID and MP6-NAT for the cobas MPX assay. As previously published [9], the infectivity conversion factors applied to the yield rates in the eight HBV infection stages change depending on the minipool size and the sensitivity of the NAT system (data presented in Appendix A). Table 1 gives the infectivity conversion factors for each of the eight infection stages and four NAT screening options. Using the ratio modelling described in Table 1, we were able to calculate the infectivity risk in the eight infection stages for the following twelve testing scenarios: HBsAg PRISM alone;HBsAg PRISM and anti-HBc;HBsAg PRISM and Ultrio Plus MP16-NAT;HBsAg PRISM, anti-HBc and Ultrio Plus MP16-NAT;HBsAg PRISM and cobas MPX MP6-NAT;HBsAg PRISM, anti-HBc and cobas MPX MP6-NAT;Ultrio Plus ID-NAT alone;Cobas MPX ID-NAT alone;HBsAg PRISM and Ultrio Plus ID-NAT;HBsAg PRISM and cobas MPX ID-NAT;HBsAg PRISM, anti-HBc and Ultrio Plus ID-NAT;HBsAg PRISM, anti-HBc and cobas MPX ID-NAT.

The regional data for infectivity risk in the eight infection stages (a–h) were calculated for four NAT testing strategies and are presented in Appendix A. The TT-HBV risk estimates in the detectable infection stages b and c were corrected for the difference in infectivity risk conversion factors for the ID and MP-NAT options (Table 1) so that the total risk without any screening test remained the same for all testing scenarios (Appendix A).

The infection risk estimates per infection stage for each of the donation categories provided the building blocks for calculating residual risk of the different testing scenarios (see results). For example, the residual risk of an HBsAg alone screening scenario is based on summing the risks in the above described eight infection stages that would not have been detected by HBsAg testing; i.e., infectivity stages a, b, d, e, g and h whereas if anti-HBc testing was also performed, then only stages a and b would be summed. If ID-NAT alone were applied, then infectivity stages a, e and h would be pertinent, whereas if ID-NAT was combined with HBsAg and anti-HBc testing, then only stage a contributes to the residual risk.

The residual risk in each of the infection/detection stages was divided by the sum of the risk in all stages to calculate the percentage risk contributed per infection stage to the overall risk if no HBV screening had been performed. Appendix A presents the same data as does Appendix A, whereby the TT-HBV risk per infection stage is expressed as a percentage of the overall risk. These proportions of the overall risk were the components for the efficacy analysis of the testing scenarios. The sum of the percent risk contribution removed in the relevant infection/detection stages a–h covered by each screening scenario was calculated. For example, HBsAg efficacy is represented by the removal of infectivity in infection/detection stages c and f, whereas dual serologic testing with HBsAg and anti-HBc would remove infectivity in stages c–h. ID-NAT alone removes infectivity in stages b, c, d, f and g whereas ID-NAT with HBsAg also eliminates infection risk by stage f.

## 3. Results

### 3.1. Relative TT-HBV Risk in Eight Infection Stages 

Figure 3 presents the modelling steps for calculating the relative TT-HBV risk in eight infection stages. Using the previously published prevalence rates for the Ultrio ID-NAT and HBsAg screened donations in four geographical regions for five infection stages, the prevalence rates for Ultrio Plus were predicted for three donation categories (first-time, lapsed plus repeat and all donations) as described in the methods and presented in Appendix A. These projected Ultrio Plus ID-NAT based prevalence rates were then used by ratio modelling to estimate TT-HBV risk per million in eight infection stages using the infectivity conversion factors calculated by the Weusten models [4,10] in Table 1. These infectivity conversion factors are equivalent to the previously published TT-HBV risk estimates expressed as a percentage of the Ultrio Plus based ID-NAT and HBsAg yield rates [9,11], which in this report are given in Appendix A. By using these infectivity conversion factors, the TT-HBV risk for each of eight infection stages and three donation categories were estimated for the ID- and MP-NAT systems of two manufacturers (Appendix A). With the use of the infectivity conversion factors in Table 1, the total TT-HBV risk without screening remained the same for the four testing strategies (Appendix A). We then calculated in Appendix A what percentage each infection stage contributed to the total risk.

The regional variation in TT-HBV risk percentages per infection stage is a function of the regional prevalence rates of the five infection stages that were detected by ID-NAT and HBsAg screening (Appendix A). Despite the variation in prevalence rates, we found for all regions a similar pattern of relative residual risk, as shown in Figure 4 for four NAT-nonreactive infection stages, i.e., the early pre-NAT WP (stage a), the second post -NAT WP (stage e), NAT-nonreactive OBI (stage h) and HBsAg-positive, HBV-DNA-negative donations (stage f), the latter to estimate the relative risk for a theoretical scenario in which these donations would be transfused. In South East Asia the relative risk posed by such HBsAg yield donations without detectable HBV-DNA was higher than in the other regions but still two-fold less than the risk of OBI transmission in ID-NAT screening setting (Figure 3, Appendix A). If however only lapsed and repeat donations are taken into account, the theoretical risk of ID-NAT-nonreactive HBsAg yield donations was more than 10-fold lower than the OBI transmission risk (and even more than 100-fold lower than the OBI transmission risk in the three other regions).

Figure 4 and Appendix A also show that the relative pre-NAT WP risk was always higher than the OBI risk in ID-NAT and MP6-NAT screening setting, but that this turned around with the least sensitive MP-NAT protocol (Ultrio Plus in MP16 configuration) in three of the four regions.

### 3.2. Residual TT-HBV Risk and Efficacy of Different Testing Scenarios

Table 2 compares the residual risk as modeled in first-time, lapsed plus repeat and all donations for twelve HBV testing scenarios. The highest predicted infection burden in the absence of any HBV testing would have occurred in South East Asia (3472 transmissions/million donations), followed by South Africa (962/million), Mediterranean Europe (334/million) and Central–North Europe (186/million). The infection risk is highest in unscreened first-time donors varying between 10,234/million in South East Asia to 1501/million in Central–North Europe. 

For example, without any HBV screening in South Africa, a transmission risk of 7888 HBV infections/million first-time donations was estimated, which diminished to 193/million in previously screened lapsed plus repeat donations. HBsAg screening reduced risk to 421/million in first-time and to 91/million in lapsed plus repeat donations. Adding Ultrio Plus ID-NAT reduced risk to 111/million in first-time and to 28/million in lapsed plus repeat donations. When all South African donations are considered together, the risk reduced from 962/million without screening to 35/million with Ultrio Plus ID-NAT alone, lower than that achieved by HBsAg and anti-HBc in combination with Ultrio Plus MP16-NAT (51/million). A similar pattern was found for the more sensitive cobas MPX HBV-NAT assay that reached a lower residual risk by itself in ID-NAT format (23/million) than when it was applied in MP6-NAT format in combination with HBsAg and anti-HBc (36/million).

In South Africa, where the relative contribution of WP donations to the overall risk was higher than in the other regions, the additional risk removed by anti-HBc testing on top of ID-NAT was very small (2/million for cobas MPX assay and 6/million for Ultrio Plus or Elite). However, in South East Asia, where the overall HBV transmission risk (3472/million) and the relative contribution of OBI was higher, the additional risk reduction by adding anti-HBc testing to ID-NAT was higher (11/million for cobas MPX and 37/million for Ultrio Plus). In all four regions, use of highly sensitive cobas ID-NAT resulted in lower residual risk than the combination of cobas MP6-NAT and anti-HBc, although the difference was only 1.2 to 1.6-fold for all donations. By contrast, the less sensitive Ultrio Plus ID-NAT option had similar residual risk as Ultrio Plus MP16-NAT in combination with anti-HBc in the European regions and provided 1.5 fold lower residual risk in one region (South Africa). In South East Asia, the impact of anti-HBc was highest, and here, the residual risk with MP16-NAT plus anti-HBc was even lower than with Ultrio Plus ID-NAT screening alone (65.7 versus 73.1 per million, respectively).

Table 3 compares the same residual risk estimates as in Table 2 but now presents these as a percentage of the overall risk without screening, whereas Table 4 gives the percentage risk removal (or the efficacy) for each of the testing scenarios, donation categories and geographical regions. For illustration the two bar diagrams of Figure 5 compare the percentage residual risk for the twelve testing strategies in all donations from South Africa and South East Asia. Although the pattern of relative risk reduction for the different testing options was similar for each of the four regions, there is variation due to differences in the relative contribution of TT-HBV risk in the different infection stages (Appendix A).

For example, in South Africa, the percent risk that remained with HBsAg testing in first-time donations was 5.3% but, in lapsed and repeat donations, was 47.4% because infections in repeat donations are limited to acute infections and to OBI NAT yields that had not been previously detected. In all South African donors, 12.1% of infectious donations are not detected by the HBsAg assay alone. Adding anti-HBc testing would reduce the undetected proportion of infectious donations to 8.2%. Adding MP-NAT to anti-HBc would further reduce the remaining risk to 3.7–5.3% and ID-NAT in combination with anti-HBc reduced the risk to 2.1–2.9%.

The most sensitive screening strategy (cobas ID-NAT, HBsAg and anti-HBc) achieved 99.2% and 90.2% efficacy in South African first-time and lapsed plus repeat donations, respectively (Table 4). In all South African donations, the efficacy of HBsAg testing alone was 87.8%; this increased to 92.4% by adding Ultrio Plus MP16-NAT, to 95.3% by adding cobas MP6-NAT and to 96.5% by adding Ultrio Plus ID-NAT. A higher efficacy of 97.7% was achieved by the most sensitive cobas ID-NAT option alone. The efficacy of the cobas ID-NAT option increased only marginally by 0.1% to 97.8% by adding anti-HBc.

The additional efficacy of anti-HBc testing relative to ID-NAT alone was greater in South East Asia (0.3% for cobas MPX and 1.07% for Ultrio Plus ID-NAT). In this region, HBsAg and cobas ID-NAT reached only marginally higher efficacy than cobas MP6-NAT in combination with HBsAg and anti-HBc (99.0% versus 98.7%). Here, Ultrio Plus ID-NAT alone had a marginally lower efficacy in all donations (97.9%) than Ultrio Plus MP16-NAT in combination with HBsAg and anti-HBc (98.1%). However, the most sensitive ID-NAT option alone (cobas MPX assay) reached again marginally higher efficacy (98.9%) than MP6-NAT with anti-HBc (98.7%) (Table 4).

The efficacy of serologic testing alone in lapsed and repeat donations was lowest in the European regions (20.3–25.0% for HBsAg and 73–74% for HBsAg and anti-HBc). In these regions, cobas ID-NAT alone reached 90.9% risk reduction in repeat and lapsed donors whereas cobas ID-NAT and anti-HBc removed up to 93.2% (Table 4). The efficacy of MP6-NAT in combination with anti-HBc in the lapsed plus repeat European donors was lower than cobas MPX ID-NAT alone (88.0–88.3% versus 90.9%). This pattern was also observed in the lapsed repeat donations of the other regions. Only in South East Asian lapsed and repeat donors was the efficacy of Ultrio Plus ID-NAT alone comparable to MP16-NAT plus anti-HBc (87.4% versus 87.2%).

## 4. Discussion

Based on our infectivity risk calculations for eight HBV infection stages using the ID-NAT screening data in four geographical regions, we reached similar conclusions as in our previous publications for efficacy of HCV and HIV screening assays in all regions [1,2]. Our models indicate that a testing scenario of highly sensitive ID-NAT alone would remove slightly more HBV transmission risk (an additional 0.22–1.35%) than MP6-NAT in combination with serologically based HBsAg and anti-HBc testing. If however a less-sensitive ID-NAT screening option without serology was compared with MP16-NAT in combination with testing for HBsAg and anti-HBc, the efficacy was 1.7% higher in South Africa but similar in Europe and even slightly lower (0.22%) in South East Asia. In this latter region, the impact of anti-HBc testing was highest because of a relatively higher prevalence of OBI- as compared to WP-NAT yield donations. The modelling results show that the impact of anti-HBc testing for removal of HBV transmission risk is highest in the scenario of MP16-NAT plus HBsAg screening (2.2–6.9% additional risk reduction by anti-HBc) and lowest with the most sensitive ID-NAT option in combination with HBsAg (0.17–0.51% additional risk reduction). 

The efficacy results of the ID-NAT testing scenarios that include HBsAg and anti-HBc also hold for an ID-NAT and anti-HBc testing strategy without HBsAg because the confirmed HBsAg-positive and ID-NAT-negative donations in our multi-regional study all tested anti-HBc-positive in supplemental testing. The relative contribution of HBsAg yield donations (that are HBV-DNA-nonreactive) to the overall risk was highest in South East Asia (0.11% and 0.32% for the two ID-NAT options in South East Asia versus 0.03–0.04% and 0.07–0.12% in the other regions). When restricted to lapsed plus repeat donations, the contribution of HBV-DNA-nonreactive HBsAg yield donations to risk reduction was minimal in three regions (0.00 to 0.02 % for cobas ID-NAT and 0.01–0.04% for Ultrio Plus ID-NAT). In South East Asia, the contribution of HBsAg on top of ID-NAT to risk reduction in the lapsed plus repeat donor group was somewhat higher (0.09 and 0.26% for the two ID-NAT options). 

A limitation of this efficacy study is that the risk estimates are totally dependent on the assumptions for the input parameters of the Weusten models (in particular, the choice of the ID_50_ in anti-HBc-negative and-positive infection stages) and the reliability of the projected Ultrio Plus ID-NAT and HBsAg screening data. Especially, donors classified as ID-NAT WP yield cases with very low viral load may not all be in the highly infectious early ramp up phase as some may represent abortive infections or have unconfirmed intermittent HBV-DNA reactivity [6]. However, the modeled relative residual risk of infectious ID-NAT-nonreactive WP and OBI donations in three of the four regions in this study (with exception of South Africa) was comparable to observational infectivity data in Japanese lookback studies [9,14,15]. To enable comparison with lookback data, we expressed the infectivity risk of detected and undetected WP and OBI donations as a percentage of the Ultrio Plus ID-NAT yield rate in these stages [9]. The OBI transmission risk was calculated to be 5.2% of the Ultrio Plus ID-NAT yield rate, which increases to 19.9% in MP16-NAT format, whereas for the more sensitive cobas MPX assay in ID-NAT and MP6-NAT configuration, the residual risk was estimated at 1.5% and 8.5% of the Ultrio Plus OBI NAT yield rate, respectively.

After reassessing the TT-HBV risk on previously published viral load distributions of South African and Egyptian HBsAg yield (HBV-DNA-nonreactive) donations [10,11] in this study (see Appendix A with erratum), we estimated that 6% of HBsAg and anti-HBc-positive donations without detectable HBV-DNA in the Ultrio Plus assay are infectious. This proportion reduces to 2.1% for the cobas MPX assay in ID-NAT format. We consider these to be worst-case estimates because theoretically the infectivity of HBsAg yield donations could be lower than that of HBV-DNA-nonreactive OBI donations. In Egyptian blood donors, a million (thousand to billion)-fold shift was observed in the ratio between noninfectious HBsAg particles and potentially infectious HBV virions when comparing the acute HBsAg and HBV-DNA-positive phase with low viral load chronic HBsAg carriers [11]. We hypothesize that neutralizing anti-preS1 antibodies play a role in this shift in HBV particle ratio and may be more present in ID-NAT-nonreactive HBsAg positive donors than in ID-NAT-nonreactive OBI donors, but for the worst-case risk analysis, in the present paper, the infectivity (ID_50_) in both infection stages was assumed to be the same (316 virions).

All risk and efficacy estimates in the present paper are calculated for RBC transfusions containing 20 mL plasma. These would be different for Fresh Frozen Plasma (FFP) transfusions; e.g., for 200 mL plasma transfusions, the OBI and HBsAg yield infectivity risk was estimated at 23% and 31% of the detected (Ultrio Plus based) yield rates, respectively. 

Finally, it must be emphasized that there is considerable uncertainty in the risk estimates in this report. The Weusten infectivity models do not give 95% confidence bounds for the risk estimates because—in addition to the considerable uncertainty and variability of ID_50_ values in stored blood components—other input parameters also have wide confidence limits, including transfusion plasma volume, the 95% and 50% LODs of the NAT method, the uncertainty in standardization of HBV-DNA genome copies or virion numbers, the viral doubling time in the acute phase, and the viral load distribution of different HBV genotypes in the later anti-HBc-positive infection stages. The infectivity-risk conversion factors for each of the infection stages in this paper represent a worst-case infectivity-risk scenario.

Given our results, it is reasonable to consider using an ID-NAT only testing scenario to replace a combined strategy of minipool NAT and serology for HBV risk reduction in repeat donors and to apply serologic testing in addition to ID NAT to first-time donors only. Furthermore, replacing MP-NAT by ID-NAT in a multiplex format with HIV and HCV, also will reduce HIV WP transmission cases that mainly have occurred in countries that used MP-NAT [16,17]. In most jurisdictions, serologic blood screening markers are mandatory, whereas molecular marker testing (NAT) is not required by regulatory agencies. However, our standardization and modelling studies over the last years provide a strong argument to no longer favor serologic above molecular testing in regulations for blood establishments. 

The completion of this efficacy analyses for removal of HBV transmission risk, combined with our previous efficacy analyses for HCV and HIV [1,2] presents us with the opportunity to use this multi-regional database for calculating the cost effectiveness of different testing scenarios for the three viruses together. An ID-NAT alone screening scenario for regular repeat donors, thereby restricting serologic testing to first-time (and lapsed) donors only, could very well turn out to be one of the most cost-effective screening scenarios. The infectivity-based risk models used in our studies [4,10] are also suitable to establish the cost effectiveness of new blood safety scenarios such as ID-NAT (or MP-NAT) in combination with pathogen inactivation.

## Figures and Tables

**Figure 1 viruses-14-02263-f001:**
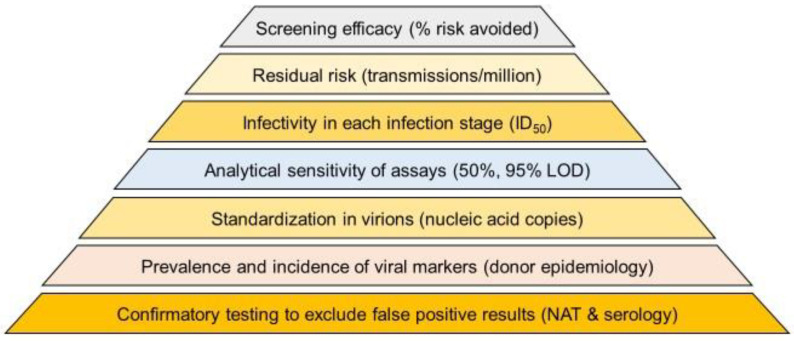
Foundations for estimating efficacy of different screening scenarios in removing viral transmission risk. The prevalence and incidence of HBV infections can only be reliably established if false-positive results in NAT and serologic assays are excluded by confirmatory testing. Since one virion in a blood component can be enough to cause transmission, it is important to standardize NAT and 50% infectious dose (in different infection stages) in virion or nucleic acid copy numbers. The residual risk of a testing scenario can then be estimated using infectivity-based mathematical models published by Weusten et al. [4,10]. Likewise the efficacy in reducing transmission risk can be estimated as a percentage relative to the baseline risk that would exist without using any screening test.

**Figure 2 viruses-14-02263-f002:**
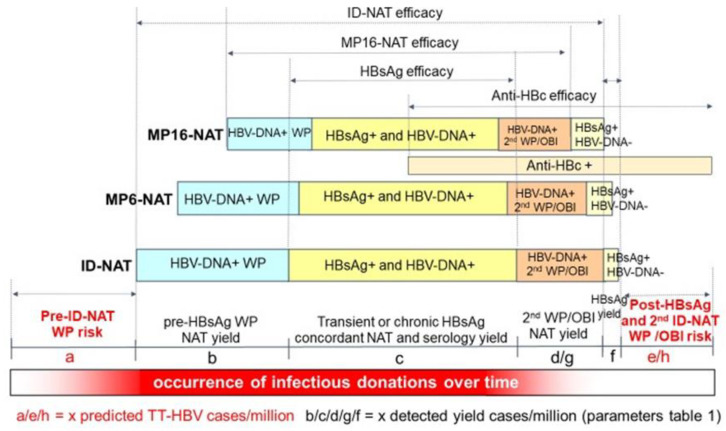
Schematic illustration of efficacy of HBV screening strategies in removing transmission risk. The intensity of the lower red colored bar on the bottom illustrates the infectivity of a blood component in different HBV infection stages. The line above this shows the evolution of HBV markers over time, resulting in different marker detection periods and non-detection (risk) periods, whereby the parameters b, c, d, g and f represent yield rates per million and the parameters a, e and h residual infection risk per million for the Ultrio Plus ID-NAT and serology testing strategy (see parameters a to h in Table 1). Moving upward, the next three bars above illustrate the proportion of HBV-transmissions interdicted by different screening assays whereby the infection risk of NAT and serology yield cases is assumed to be 100% in stages b and c but less in the other stages (see Table 1). The dotted line arrows indicate the efficacy in risk removal by different screening tests.

**Figure 3 viruses-14-02263-f003:**
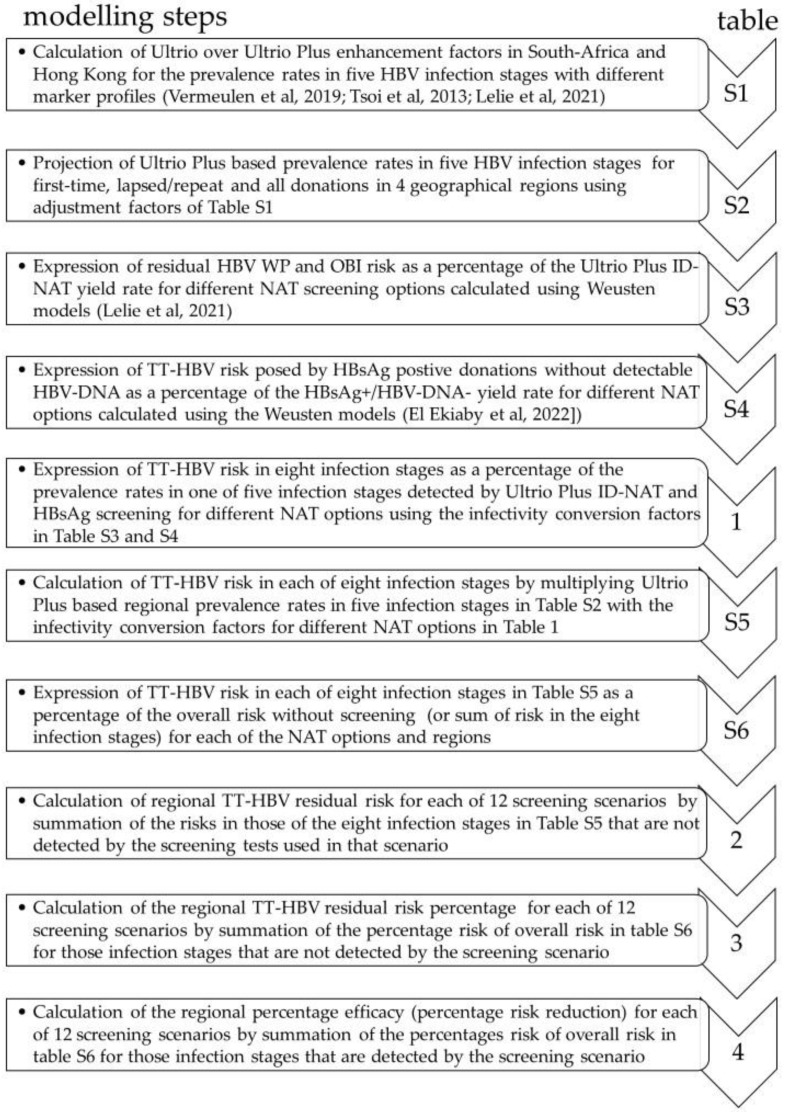
Subsequent modelling steps to estimate the efficacy of different testing scenarios in removing TT-HBV risk using screening data from four geographical regions. Each of the modelling steps is documented with a table in this report (i.e., Appendix A and Table 1, Table 2, Table 3 and Table 4) [7,8,9,11].

**Figure 4 viruses-14-02263-f004:**
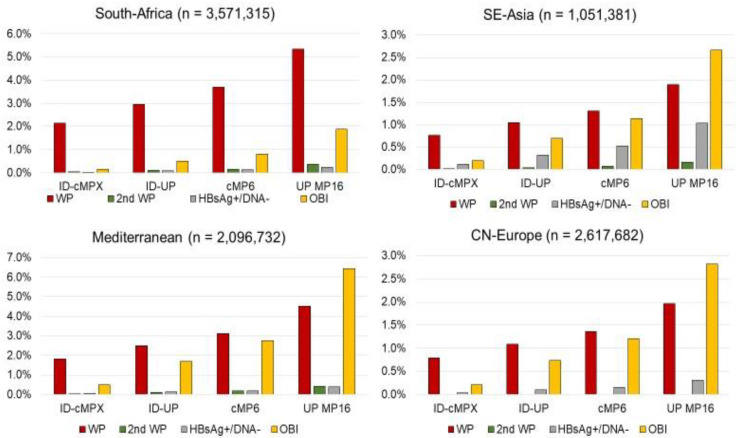
Residual TT-HBV risk for RBC transfusions expressed as a percentage of the overall risk in the scenario where no HBV screening had been performed. Each bar diagram represents residual risk percentages in one geographical region for 4 NAT screening scenarios (ID-cMPX = individual donation NAT with cobas MPX assay; ID-UP = individual donation NAT with Ultrio Plus assay; cMP6 = cobas MPX assay tests on minipools of 6 donations; UP MP16 = Ultrio Plus tests on minipools of 16 donations). Residual risk percentages were calculated from screening data on all donations in each of the four geographical regions. For each screening scenario, residual risk percentages are shown for four infection stages, i.e., from left to right: early pre-NAT WP (brown bar): late acute post-NAT WP (green bar), NAT-nonreactive HBsAg-positive (grey bar) and NAT-nonreactive OBI (orange bar).

**Figure 5 viruses-14-02263-f005:**
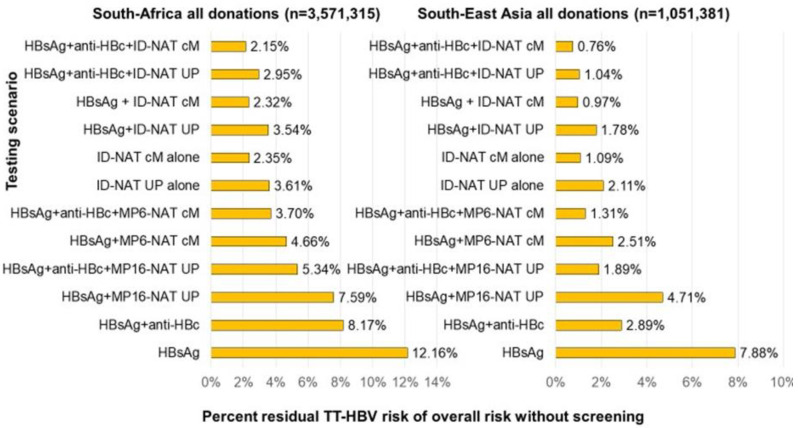
Residual TT-HBV risk for RBC transfusions based on screening data in two of the four geographical regions estimated for 12 testing scenarios (UP = Ultrio Plus, cM = cobas MPX, MP6 = minipools of 6; MP16 = minipools of 16) and expressed as a percentage of the overall risk in case no HBV testing had been performed.

**Table 1 viruses-14-02263-t001:** RBC transfusion-infectivity risk conversion factors for eight HBV infection stages and four NAT systems calculated by Weusten risk models [4,10] from the yield rates detected by Ultrio Plus ID-NAT and serology. The same infectivity factors are presented as a percentage of the WP and OBI ID-NAT yield rates in Appendix A, previously published as Table 4 of a recent publication of this multi-center study [9].

HBV Infection Stage	Yield Rate/Million Ultrio Plus ID-NAT	Infection Risk/Million Ultrio Plus ID-NAT	Infectivity Risk (Predicted TT-HBV Cases/Million)			
			Ultrio Plus ID-NAT	Cobas MPX ID-NAT	Cobas MPX MP6-NAT	Ultrio Plus MP16-NAT
Pre-NAT WP undetected		a	0.565b	0.412b	0.709b	1.023b
Pre-HBsAg WP NAT yield	b		b	b + (0.565–0.412)b	b + (0.565–0.709)b	b + (0.565–1.023)b
HBsAg + /DNA + yield	c		c	c + (0.060–0.021)f	c + (0.060–0.096)f	c + (0.060–0.190)f
Post-HBsAg WP NAT yield	d		0.30d	0.337d	0.267d	0.153d
Post-NAT WP undetected		e	0.052d	0.015d	0.085d	0.199d
HBsAg + /DNA-yield	f		0.060f	0.021f	0.096f	0.190f
OBI NAT yield	g		0.30g	0.337g	0.267g	0.153g
OBI undetected		h	0.052g	0.015g	0.085g	0.199g

**Table 2 viruses-14-02263-t002:** Residual HBV transmission risk/million for different screening options and donation categories as calculated from yield rates in four geographical regions using Weusten risk models [4,10] for RBC transfusion.

Scheme.	South Africa			South East Asia			Mediterranean			Central–North Europe		
	First-Time	Lpsd + Rpt	All	First-Time	Lpsd + Rpt	All	First-Time	Lpsd + Rpt	All	First-Time	Lpsd + Rpt	All
no screening	7887.8	192.7	962.4	10232.7	433.4	3471.6	2004.6	74.3	333.8	1505.9	20.0	186.3
HBsAg	421.2	91.4	117.0	307.6	240.6	273.5	124.5	59.3	63.5	21.2	15.0	14.9
HBsAg + a-HBc	230.1	68.5	78.6	98.6	85.0	100.5	66.0	19.2	23.1	10.8	5.3	5.6
HBsAg + MP16-UP	258.5	57.7	73.1	182.6	143.5	163.5	76.2	35.2	37.9	12.9	9.0	8.9
HBsAg + a-HBc + MP16-UP	150.4	44.8	51.4	64.5	55.6	65.7	43.2	12.5	15.1	7.1	3.5	3.7
HBsAg + MP6-cM	150.4	36.6	44.9	95.1	76.1	87.3	44.0	18.4	20.2	7.4	4.8	4.8
HBsAg + a-HBc + MP6-cM	104.2	31.1	35.6	44.7	38.5	45.5	29.9	8.7	10.4	4.9	2.4	2.5
ID-UP alone	118.2	28.1	34.8	100.4	54.8	73.1	35.3	12.9	14.7	6.9	3.4	3.6
ID-cM alone	71.1	19.0	22.6	46.8	29.4	37.8	20.9	6.8	7.9	3.8	1.8	1.9
HBsAg + ID-UP	111.3	28.1	34.1	66.5	53.7	61.8	32.5	12.8	14.3	5.4	3.4	3.4
HBsAg + ID-cM	68.7	19.0	22.3	34.9	29.0	33.8	19.9	6.8	7.8	3.3	1.8	1.9
HBsAg + a-HBc + ID-UP	83.1	24.7	28.4	35.6	30.7	36.3	23.8	6.9	8.3	3.9	1.9	2.0
HBsAg + a-HBc + ID-cM	60.6	18.0	20.7	26.0	22.4	26.5	17.4	5.0	6.1	2.9	1.4	1.5

UP = Ultrio Plus, cM = cobas MPX.

**Table 3 viruses-14-02263-t003:** Residual HBV transmission risk for different screening options and donation categories as calculated from yield rates in four geographical regions using Weusten risk models [4,10] for RBC transfusion expressed as a percentage of infection risk without screening.

Screening Scenario	South Africa			South East Asia			Mediterranean			Central–North Europe		
	First-Time	Lpsd + Rpt	All	First-Time	Lpsd + Rpt	All	First-Time	Lpsd + Rpt	All	First-Time	Lpsd + Rpt	All
no screening	100	100	100	100	100	100	100	100	100	100	100	100
HBsAg	5.34	47.42	12.16	3.01	55.52	7.88	6.21	79.72	19.03	1.41	75.04	7.98
HBsAg + a-HBc	2.92	35.58	8.17	0.96	19.62	2.89	3.29	25.78	6.91	0.72	26.57	3.00
HBsAg + MP16-UP	3.28	29.95	7.59	1.78	33.12	4.71	3.80	47.35	11.37	0.86	44.77	4.78
HBsAg + a-HBc + MP16-UP	1.91	23.26	5.34	0.63	12.82	1.89	2.15	16.85	4.51	0.47	17.37	1.96
HBsAg + MP6-cM	1.91	18.98	4.66	0.93	17.56	2.51	2.20	24.71	6.06	0.49	23.74	2.56
HBsAg + a-HBc + MP6-cM	1.32	16.12	3.70	0.44	8.89	1.31	1.49	11.68	3.13	0.33	12.04	1.36
ID-UP alone	1.50	14.60	3.61	0.98	12.65	2.11	1.76	17.31	4.41	0.46	16.81	1.92
ID-cM alone	0.90	9.87	2.35	0.46	6.79	1.09	1.04	9.10	2.38	0.25	9.08	1.04
HBsAg + ID-UP	1.41	14.59	3.54	0.65	12.39	1.78	1.62	17.28	4.28	0.36	16.75	1.82
HBsAg + ID-cM	0.87	9.87	2.32	0.34	6.69	0.97	0.99	9.08	2.33	0.22	9.06	1.00
HBsAg + a-HBc + ID-UP	1.05	12.84	2.95	0.35	7.08	1.04	1.19	9.31	2.49	0.26	9.59	1.08
HBsAg + a-HBc + ID-cM	0.77	9.37	2.15	0.25	5.16	0.76	0.87	6.79	1.82	0.19	6.99	0.79

UP = Ultrio Plus, cM = cobas MPX.

**Table 4 viruses-14-02263-t004:** Efficacy of screening scenarios in removing HBV transmission risk for different donation categories calculated by Weusten models [4,10] for RBC transfusions from residual risk percentages in four geographical regions (Table 3).

Screening Scenario	South Africa			South East Asia			Mediterranean			Central–North Europe		
	First-Time	Lpsd + Rpt	All	First-Time	Lpsd + Rpt	All	First-Time	Lpsd + Rpt	All	First-Time	Lpsd + Rpt	All
No screening	0.0	0.0	0.0	0.0	0.0	0.0	0.0	0.0	0.0	0.0	0.0	0.0
HBsAg	94.7	52.6	87.8	97.0	44.5	92.1	93.8	20.3	81.0	98.6	25.0	92.0
HBsAg + a-HBc	97.1	64.4	91.8	99.0	80.4	97.1	96.7	74.2	93.1	99.3	73.4	97.0
HBsAg + MP16-UP	96.7	70.0	92.4	98.2	66.9	95.3	96.2	52.7	88.6	99.1	55.2	95.2
HBsAg + a-HBc + MP16-UP	98.1	76.7	94.7	99.4	87.2	98.1	97.8	83.2	95.5	99.5	82.6	98.0
HBsAg + MP6-cM	98.1	81.0	95.3	99.1	82.4	97.5	97.8	75.3	93.9	99.5	76.3	97.4
HBsAg + a-HBc + MP6-cM	98.7	83.9	96.3	99.6	91.1	98.7	98.5	88.3	96.9	99.7	88.0	98.6
ID-UP alone	98.5	85.4	96.4	99.0	87.4	97.9	98.2	82.7	95.6	99.5	83.2	98.1
ID-cM alone	99.1	90.1	97.7	99.5	93.2	98.9	99.0	90.9	97.6	99.7	90.9	99.0
HBsAg + ID-UP	98.6	85.4	96.5	99.4	87.6	98.2	98.4	82.7	95.7	99.6	83.2	98.2
HBsAg + ID-cM	99.1	90.1	97.7	99.7	93.3	99.0	99.0	90.9	97.7	99.8	90.9	99.0
HBsAg + a-HBc + ID-UP	98.9	87.2	97.1	99.7	92.9	99.0	98.8	90.7	97.5	99.7	90.4	98.9
HBsAg + a-HBc + ID-cM	99.2	90.6	97.8	99.7	94.8	99.2	99.1	93.2	98.2	99.8	93.0	99.2

UP = Ultrio Plus, cM = cobas MPX.

## Data Availability

All data are reported in the present manuscript and Appendix A.

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
