# Peer review of "Efficacy of Different Testing Scenarios in Reducing Transfusion-Transmitted Hepatitis B Virus (TT-HBV) Infection Risk"

_viruses, 2022, doi:10.3390/v14102263_

Round 1
Reviewer 1 Report
This analysis builds on previous work by the authors on hepatitis B (HBV) residual risk. An important strength is the application of data from multiple countries, some of which have higher incidence. The objective is to estimate the residual risk over eight infectivity stages of HBV, and evaluate risk reduction of various testing algorithms. Results suggest that ID NAT gives very good risk reduction, mini-pool NAT not quite so much but there may be some benefit from HBsAg and anti-HBc depending on the infection stage profile of a region especially if using MP NAT.
Specific comments:
1. The paper is very complicated to read. It would be easier to follow with a schematic diagram showing the reader which information was used or generated where in the process and identifying which table for each.
2. The results are presented as the percentage reduction in residual risk of various testing strategies relative to no testing at all. The baseline residual risk with no testing should be the same for first time and lapsed/repeat because repeat risk only goes down after testing culls out most non-incident infections. I am guessing that the lapsed/repeat risk without testing used for the analysis is actually the lower risk in repeat donors screened with some other algorithm in the countries that provided the data. It would make more sense to report just the total residual risk reduction relative to the real untested residual risk and leave out the first time and lapsed/repeat.
3. The analysis highlights variability in infection stage profile by region which donor testing policy must address. This variability is a reflection of epidemiology, which is very much influenced by government public health and immigration policy. The WHO has set a target for elimination (or substantive reduction) of HBV by 2030. As Viruses has a wide readership, the authors thoughts on how their analysis may be helpful in setting donor testing policy with declining incidence would be of interest.
4. It would also be helpful to readers to set the stage for donor policy in the introduction, since not all have background in blood donation. Specifically, that all donors are screened by questionnaire for risk factors and have testing of some sort currently in place. The discussion should explain why risk needs to be so very low for transfusion. People who do not work in transfusion sometimes find the obsession with lowering small risks confusing.
Author Response
Comments and Suggestions for Authors (reviewer 1)
This analysis builds on previous work by the authors on hepatitis B (HBV) residual risk. An important strength is the application of data from multiple countries, some of which have higher incidence. The objective is to estimate the residual risk over eight infectivity stages of HBV, and evaluate risk reduction of various testing algorithms. Results suggest that ID NAT gives very good risk reduction, mini-pool NAT not quite so much but there may be some benefit from HBsAg and anti-HBc depending on the infection stage profile of a region especially if using MP NAT.
Specific comments:
- The paper is very complicated to read. It would be easier to follow with a schematic diagram showing the reader which information was used or generated where in the process and identifying which table for each.
Response
This is a very good suggestion. We have added figure 3 showing what calculations were made in the subsequent modeling steps, whereby for each step the table number is given in which the modeling results are documented, i.e., in six supplemental tables and 4 regular tables. Hopefully the addition of this figure and further clarifications in the text makes the paper more comprehensible (see track changes).
- The results are presented as the percentage reduction in residual risk of various testing strategies relative to no testing at all. The baseline residual risk with no testing should be the same for first time and lapsed/repeat because repeat risk only goes down after testing culls out most non-incident infections. I am guessing that the lapsed/repeat risk without testing used for the analysis is actually the lower risk in repeat donors screened with some other algorithm in the countries that provided the data. It would make more sense to report just the total residual risk reduction relative to the real untested residual risk and leave out the first time and lapsed/repeat.
Response
The screening of first time donors removes most of the chronic infections (HBsAg and occult carriers) and some acute infections. In the subsequent donations of those that become regular repeat donors, in addition to interdiction of acute infections (and deferral of the acutely infected donors) chronic infections that were previously missed because of low viral load or fluctuating HBV-DNA and HBsAg near the detection limit can be detected and donors deferred (since in this multi-center study anti-HBc was not used as a screening test). Indeed, the residual WP or OBI risk could be the same in the first time and repeat donors if such culling had not already occurred prior to this study and if the underlying demographics and risk behaviors are the same in both donor groups. However, since the vast majority of chronic infections including occult HBV carriers had been removed in the repeat donor group, the overall risks in the prescreened populations are significantly lower than would have been the case had no screening of previous donations been performed. We reported the risk analysis using the screening data for first time and lapsed/repeat donors separately (and not only for all donations in aggregate) mainly to examine the contribution of TT-HBV risk posed by HBsAg positive donors without detectable HBV-DNA. As our analysis shows this contribution of HBsAg+/HBV-DNA- yield donors to the overall risk is low in first time donors but negligible in repeat donors. This is why an ID-NAT only testing strategy could be considered for the repeat donors and anti-HBc/HBsAg testing is still recommended for first time donors only.
- The analysis highlights variability in infection stage profile by region which donor testing policy must address. This variability is a reflection of epidemiology, which is very much influenced by government public health and immigration policy. The WHO has set a target for elimination (or substantive reduction) of HBV by 2030. As Viruses has a wide readership, the authors thoughts on how their analysis may be helpful in setting donor testing policy with declining incidence would be of interest.
Response
Although there is regional variation in the relative proportion of HBV infected donors in the eight infection stages, the patterns of risk reduction by the different testing strategies are similar. We decided not to discuss a future scenario of elimination of HBV infection (by universal vaccination and potential, but still unsuccessful, development of direct acting anti-viral drugs similar to those now available for HCV) in our manuscript and only base our conclusions on the available regional ID-NAT screening data of approximately a decade ago. One can expect that in the future more and more blood donors are vaccinated against HBV in their childhood so that chronic infections in first time donors will further be reduced over time and the relative proportion of vaccine breakthrough or abortive infections among acute HBV infections will increase. In this setting the efficacy of HBsAg testing becomes even less, and HBV screening will be more dependent on sensitive HBV-DNA screening to identify potential infectious breakthrough or abortive infections with very low viral loads.
- It would also be helpful to readers to set the stage for donor policy in the introduction, since not all have background in blood donation. Specifically, that all donors are screened by questionnaire for risk factors and have testing of some sort currently in place. The discussion should explain why risk needs to be so very low for transfusion. People who do not work in transfusion sometimes find the obsession with lowering small risks confusing.
Response
We appreciate the recommendation that more background on donor screening policies and acceptable levels of residual risk in the blood transfusion setting may be helpful for the general readers of Viruses. However, due to length limitations we are reluctant to discuss this broader context as this departs from our focus on the risk reduction achieved by the currently used testing algorithms. Furthermore, this background subject matter is extensively presented in our earlier cited manuscripts.
Reviewer 2 Report
1. Adjustment factors are important in calculations. The authors need to explain in the methods why the adjustment factors found in South Africa could use to those from Mediterranean Europe and Central-North Europe.
2. Table 1, the infectivity risk of Ultrio Plus MP16-NAT in Pre-HBsAg WP NAT yield is b+(0.0565-1.023)b or b+(0.565-1.023)b? Please confirm.
3. The order of tables is confusing. I suggest to combine Table 2,3 and 4 into one table. The results can be present as “Residual HBV transmission risk/million (Residual HBV transmission risk%)” or “Removing HBV transmission risk/million (Removing HBV transmission risk%)”.
4. In page 6, What do the numbers at the end of sentences a through h mean? Are they reference numbers?
Author Response
Comments and Suggestions for Authors (reviewer 2)
- Adjustment factors are important in calculations. The authors need to explain in the methods why the adjustment factors found in South Africa could use to those from Mediterranean Europe and Central-North Europe.
Response
We agree and have added more background to the methods section explaining why we could not simply use the Weusten model for the original Ultrio assay that had inconsistent sensitivity for different HBV samples. Although the reason for this deficiency of the older and no longer available Ultrio assay is not exclusively related to the HBV genotypes, it likely is a substantial factor since genotypes may differ with respect to the proportion of samples with longer double stranded portions of the HBV genome that became detectable with the alkalic shock modifications introduced in the Ultrip Plus and Ultrio Elite assays. We therefore mentioned the predominance of HBV genotype A genotypes in South Africa and Central-North Europe to justify this choice but also mentioned the contribution of HBV genotype D infections that are more often found in the Mediterranean region. In the discussion we mentioned the limitations of using the projections for the Ultrio Plus based prevalence rates in general. We also mentioned in the methods that we had chosen the South African adjustment factors for the European regions because they were lower. It could very well be that they are somewhat higher for genotype D infections, but this would not change the general conclusions of this paper which for the most part are based on the South-African and South-East Asian data.
- Table 1, the infectivity risk of Ultrio Plus MP16-NAT in Pre-HBsAg WP NAT yield is b+(0.0565-1.023)b or b+(0.565-1.023)b? Please confirm.
Response
This was indeed a typing error. Thanks for having noticed this. The formula is b+(0.565-1.023)b
- The order of tables is confusing. I suggest to combine Table 2,3 and 4 into one table. The results can be present as “Residual HBV transmission risk/million (Residual HBV transmission risk%)” or “Removing HBV transmission risk/million (Removing HBV transmission risk%)”.
Response
We have included a figure that shows the modelling steps and corresponding tables with the calculated results. We will also ask the editors to present these three related final tables in close proximity to each other. We believe this will be more effective than combining so much data into one very long table.
- In page 6, What do the numbers at the end of sentences a through h mean? Are they reference numbers?
Response
Yes these are literature reference numbers and should have been in superscript format, which has been corrected.